# Estimating the potato farming efficiency: A comparative study between stochastic frontier analysis and data envelopment analysis

**Shamima Sultana**[1]\*, **Md. Moyazzem Hossain**[2]\*, **Md. Nurul Haque**[3]

**1** Department of Economics, Jagannath University, Dhaka, Bangladesh, **2** Department of Statistics, Jahangirnagar University, Dhaka, Bangladesh, **3** Department of Economics, Jahangirnagar University, Dhaka, Bangladesh

\* shamima@eco.jnu.ac.bd (SS); hossainmm@juniv.edu (MMH)

## Abstract

### Background

The government of Bangladesh has been trying to encourage potato consumption to reduce pressure on rice consumption and earn foreign currency along with ensuring zero hunger that helps to achieve the Sustainable Development Goal. It is necessary to use farmers' resources and current technology more efficiently to meet the demand. Therefore, the authors aimed to evaluate the farm-level efficiency of potato farming in Bangladesh.

### Methods and materials

The Cobb-Douglas Stochastic Frontier Analysis (SFA) and the input-oriented Data Envelopment Analysis (DEA) methods are used to compute farm-level technical, allocative, and economic efficiencies and inefficiency of potato farming. The primary data were collected through interviews of 300 potato farmers from Munshigonj, Rangpur, Dinajpur, and Joypurhat districts of Bangladesh.

### Results

The findings revealed that the efficiency score of the SFA model is higher than the DEA model, which implies that the SFA frontier fits better than the DEA frontier. In the case of DEA, variable returns to scale (VRS) technical efficiency (TE) enveloped data more closely than constant returns to scale (CRS) TE. Results of efficiency suggest significant economic, technical, and allocative inefficiencies in potato farming and there is a scope to increase potato production levels through efficiency improvement. Inefficiency analysis shows that infrastructure and socio-economic factors jointly influence potato production variability.

### Conclusions

The authors suggest for using the SFA to find efficiencies in the agriculture sector. To achieve efficiency in potato production, the government needs to pay attention for improving

**Data Availability Statement:** All relevant data are within the paper and its Supporting information files.

**Funding:** The author(s) received no specific funding for this work.

**Competing interests:** The authors have declared that no competing interests exist.

the allocative and economic efficiencies along with emphasizing to choose the appropriate technology and efficient use of resources for the scale of operation.

## Introduction

The potato (Solanum tuberosum L.) is the most productive crop and the third-largest source of food on earth [1–3]. Despite being known as a rice-eating country, Bangladesh produces and consumes a lot of potatoes each year and it has steadily grown in popularity. Both rich and poor people use potatoes as a food crop and a vegetable. People eat cooked potatoes, fries, and chips [4]. After rice and wheat, potatoes are one of Bangladesh's key food crops. Over the past five years, the nation has produced potatoes at a rate of 10 million MT on average. Bangladesh is the fourth-largest producer of potatoes in Asia and is in the top 15 in the world [5]. The geographical suitability of Bangladesh is also responsible for the increased production of potatoes in every year [6]. However, they are vulnerable to climate change since the production of potatoes is extremely sensitive to a variety of abiotic factors, such as temperature and soil salinity [1, 7].

Nowadays, developing countries' agricultural farming efficiency has been a topic of considerable interest in development literature. Efficiency is a performance measure and success indicator. The efficiency studies show the possibility of productivity raising through efficiency improvement without introducing new technologies or increasing the resource base. Farming efficiency of the agriculture sector is considered an essential factor in balancing population growth and agricultural output (food and raw materials). Inefficiency estimations help in taking a decision to raise productivity, whether through efficiency improvement or the introduction of new technology. In Bangladesh, potatoes have become a cash crop, and their importance is rising rapidly in the domestic and international markets. A previous study highlighted that ineffective transportation, inadequate storage, a shortage of capital, farmers' ignorance of market prices, illiteracy, and the syndicate system of middlemen are some of the major contributing causes to Bangladesh's ineffective potato marketing system [8]. Farmers' farming efficiency is an important tool for optimizing production decisions and strengthening the farms' capacity to face increasing input costs, changing market conditions, rapid technological progress, and economic hardships to reach optimum output levels. Farmers' farming efficiency depends on their level of education, years of experience, land fragmentation, use of seeds, modern technology, fertilizer, and other inputs [9–16]. It is necessary to undertake research efforts to estimate the efficiency of potato farming to ensure better ways and sustainably improve potato productivity. Farmers need to improve their farming efficiency to achieve financial success and profit. There are variations in the resources used effectively in different farms. Resources should be utilized optimally to reach the maximum level of output and income. Farms' decision about raising farming efficiency depends on identifying inefficiency factors. Therefore, policymakers may give more emphasis on the improvement of efficiency. It is important to assess the efficiency of potato farming to find out the potential and possibilities for the expansion and sustainability of potato production in Bangladesh.

There is little information about the economic, technical, and allocative efficiencies of the potato sector in Bangladesh. Most studies estimate technical efficiency using SFA, however, some studies estimate economic, technical, and allocative efficiencies in the agriculture sector of Bangladesh using DEA and SFA [17–20]. There are several studies on the comparison between DEA and SFA for different crops in different countries [9, 21–30]. To the best of our knowledge, in the context of Bangladesh, there is a gap in research on potato farming efficiency

using both SFA and DEA. The study's objectives are to estimate the potato farming efficiency (economic, technical, and allocative efficiencies), identify the explanatory factors of potato farming efficiency, and find any significant difference between the efficiency results of SFA and DEA methods. This study emphasizes for utilizing the allocated resources efficiently in potato farming which may help to improve the prevailing situation through developing policy parameters.

## Methods and materials

### Data and variables

This study collected data from 300 individual farmers using a semi-structured questionnaire from four districts Munshigonj, Rangpur, Dinajpur, and Joypurhat of Bangladesh which is commercial potato-grown areas. Before participating in the survey, the authors explained the purpose of the study to the participants and assured them that their answers would be kept private, that no personally identifiable information would be shared, and that their verbal consent would be obtained. This study collected the following variables from the survey: land (decimal), yield (kg), Seed (kg), tilling (Tk.), cost of labor (Tk.), irrigation (Tk.), fertilizer (Tk.), vitamin (Tk.), and pesticides (Tk.), in thousand Tk. (Tk. is the currency of Bangladesh). Inefficiency factors such as education (year), age (year), experience (year), training (dummy), access to credit (dummy), land fragmentation (average plot size), weed uprooting cost (Tk.), household size (number of family members), and cold storage facility (dummy) are used in this study.

### Ethics statement

Before participating in the survey, the authors explained the purpose of the study to the participants and assured them that their answers would be kept private, that no personally identifiable information would be shared, and that their verbal consent would be obtained. This study do not take any personal identifiable information and sample from human body.

### Methods of estimation

Farrell's (1957) pioneering article on efficiency measurement helps develop different efficiency analysis methods. SFA and DEA are mostly used in efficiency-related research work worldwide [31]. Generally, SFA is used to assess the performance of the agricultural sector. DEA is used to estimate the efficiency of the non-agricultural sector, e.g., public utilities, banks, hospitals, education institutions, etc. However, the number of research carried out using DEA and SFA methods for crops. This research employs the SFA and DEA methods to estimate two districts' efficient and inefficient determinants. Two methods are discussed as follows:

### Stochastic Frontier Analysis (SFA)

Aigner, et al. (1977) and Meeusen and van den Broeck (1977) independently suggested the stochastic frontier function [32, 33]. The Cobb-Douglas type production function has been used extensively by researchers worldwide and is suited for the examination of applied research in different fields including industry, agricultural production, and so on [22, 34–38]. The existing literature motivates us to use the self-dual Cobb-Douglas SFA Method in this study. It needs to assume a functional form for the production technology and the distribution of the technical inefficiency term. The economic, technical, and allocative efficiencies are computed from it's dual cost frontier model. The Cobb-Douglas stochastic frontier specified for this study is as

follows:

$$\ln \tilde{y}_i = \beta_0 + \sum_{k=1}^{8} \ln x_{ik} \ln x_{ik} + v_i - u_i, \ i = 1, \ 2, \ldots, 300 \ \text{number of farms} \tag{1}$$

where ln: the natural logarithm, $x_1$: land, $x_2$: labor, $x_3$: tilling, $x_4$: seed, $x_5$: fertilizer, $x_6$: irrigation, $x_7$: pesticide, $x_8$: vitamin, $\beta_0$: technical efficiency level, $\beta_1, \ldots .\beta_8$: coefficients of inputs concerning output level, $v_i$: the random error that farmers cannot control (weather, pests, diseases, measurement error in the output variable, etc.), $u_i$: the non-negative random error measures the technical inefficiency relative to the stochastic frontier, $\tilde{y}_i$: the farms observed output adjusted for the stochastic random noise captured by $v_i$. For $u_i = 0$ the farm lies on the Stochastic Production Function and $u_i > 0$ represents inefficiency in the farm. It is assumed that $Cov$ $(v_i, u_i) = 0$, $Cov(v_i, x_i) = 0$, and $Cov(u_i, x_i) = 0$. The variance parameters of the model are expressed as: $\sigma^2 = \sigma_v^2 + \sigma_u^2$, $\gamma = \frac{\sigma_u^2}{\sigma^2}$, $0 \leq \gamma \leq 1$, where, $\gamma = 0$: absence of stochastic technical inefficiency when stochastic frontier model becomes a average frontier model, and $\gamma = 1$: the stochastic random error term is absent when the stochastic frontier model becomes a full frontier model [39]. The parameter $\lambda = \frac{\sigma_u}{\sigma_v}$ is a measure of the comparative variability of two inefficiency sources and $\lambda^2 \to 0$ implies that $\sigma_v^2 \to \infty$ and/or $\sigma_u^2 \to 0$ means that the random shocks dominate in explaining the inefficiency. When $\sigma_v^2 \to 0$ then, the gaps to the frontier are due to technical inefficiency.

The $i^{\text{th}}$ farm-specific technical efficiency is the ratio of $y_i$ and $y^*$, given the input levels and can be written as $TE_i = \frac{y_i}{y_i^*} = \frac{f(x_i, \beta)e^{v_i - u_i}}{f(x_i, \beta)e^{v_i}} = e^{-u_i}$, $0 \leq TE_i \leq 1$, where $y_i$ is the observed output of $i^{\text{th}}$ farm, $y^*$ is the corresponding frontier output, $u_i$ are non-negative truncations of the $N(\mu, \sigma_u^2)$ distribution and $\mu = z_i \delta_i$ where $z_i$ is a $(k \times 1)$ vector of variables that influence efficiency and $\delta_i$ to be estimated $(1 \times k)$ vector of parameters.

Eq (1) constitutes the basis for obtaining the technically efficient input vector $x_{ik}^T$. The dual stochastic frontier cost function model is analytically derived from the stochastic production model. The economically efficient input vector $x_{ik}^T$ is derived from the dual stochastic frontier cost function. The dual stochastic frontier cost function model is

$$C(p_{ik}, \tilde{y}_i) = \alpha_0 \prod_{k=1}^{8} p_{ik}^{\beta_{ik}\alpha_{ik}} \tilde{y}_i^{\alpha_{ik}} \tag{2}$$

where, $C(p_{ik}, \tilde{y}_i)$ is the cost function, $\alpha_0 = \left( \frac{1}{\beta_0^{\alpha_{ik}}} \right) \left( \dfrac{\sum\limits_{k=1}^{8} \beta_{ik}}{\prod\limits_{k=1}^{8} \beta_{ik}^{\beta_{ik}\alpha_{ik}}} \right)$ and $\alpha_{ik} = \dfrac{1}{\sum\limits_{k=1}^{8} \beta_{ik}}$.

Differentiating (2) with respect to each input's price and applying Shephard lemma the system of input demand function can be written as:

$$x_{ik}^E = \frac{\partial C(p_{ik}, \tilde{y}_i)}{\partial p_{ik}} = \alpha_0 (\beta_{ik} \alpha_{ik}) \prod_{k=1}^{8} \frac{1}{p_{ik}} p_{ik}^{\beta_{ik}\alpha_{ik}} \tilde{y}_i^{\alpha_{ik}}. \tag{3}$$

From the result of stochastic frontier production function (1), we can get the technically efficient input vector $x_{ik}^T$. Multiplying the observed input vector $x_{ik}$, the technically efficient input vector $x_{ik}^T$ and the economically efficient input vector $x_{ik}^E$ by the input price vector provides the observed, technically efficient, and economically efficient costs of production of the i-th farm equal to $p_{ik}x_{ik}$, $p_{ik}x_{ik}^T$, and $p_{ik}x_{ik}^E$ respectively which compute the TE, AE, and EE

indices for the ith farm as:

$$TE = \frac{p_{ik}x_{ik}^T}{p_{ik}x_{ik}}, \; AE = \frac{p_{ik}x_{ik}^E}{p_{ik}x_{ik}^T}, \; \text{and } EE = \frac{p_{ik}x_{ik}^E}{p_{ik}x_{ik}}.$$

## Data Envelopment Analysis (DEA) model

The DEA is a linear programming method used to formulate a piece-wise linear surface over the input and output data points. The linear programming problems are solved for each farm in the sample to construct the frontier surface and produce the level of inefficiency. The level of inefficiency is the gap between the frontier and the observed data point. The efficient farm with an efficiency score of one lies on the production frontier, and the inefficient farm with an efficiency score of less than one exists beneath the frontier. Charnes et al., (1978) proposed an input orientation DEA method that assumed constant returns to scale (CRS) [40]. Moreover, Banker et al., (1984) assumed variable returns to scale (VRS) in the DEA model [41]. Since this study considers potatoes, a single output and eight inputs it selects the input orientation model. This model asses how much input can be changed proportionally to produce a fixed amount of output.

## Determinants of inefficiency

The following equation presents the inefficiency effects model. For the SFA method, technical efficiency is calculated in a single-stage method in which the technical inefficiency effects are modeled as a function of socio-economic characteristics and infrastructure factors. The DEA method estimates inefficient effects using the Tobit regression model.

$$IE_i = \delta_0 + \delta_1 z_{i1} + \delta_2 z_{i2} + \delta_3 z_{i3} + \delta_4 z_{i4} + \delta_5 z_{i5} + \delta_6 z_{i6} + \delta_7 z_{i7} + \delta_8 z_{i8} + \delta_9 z_{i9} + w_i \qquad (4)$$

where, $IE_i$: inefficiency, $z_1$: age (year), $z_2$: education (years of schooling), $z_3$: experience (year), $z_4$: land fragmentation (number of plots), $z_5$: family size, $z_6$: deweeding, $z_7$: access to credit (dummy), $z_8$: cold storage (dummy), $z_9$: training (dummy), and $w_i$: error term.

## Results and discussion

The summary statistics of the output and input variables as well as variables that caused technical inefficiency of the potato farming used in this study are presented in Table 1. Results revealed that on an average 26.39 thousand kg of potatoes were produced by the farmers with a minimum of 0.28 thousand kg and a maximum of 378 thousand kg. It is observed that the minimum and maximum cost for harvested land for potatoes were 0.47 thousand Tk. and 270 thousand Tk. respectively. Among the expenses, seed cast was the most expensive. In some areas of Bangladesh, the farmers used less irrigation for potatoes resulting in the minimum cost for this purpose being 0.18 thousand Tk. Moreover, labor cost was the second largest expenditure sector with an average of 40.99 thousand Tk. with a minimum of 1.20 thousand Tk. and a maximum of 539 thousand Tk.

The findings depict that the average age of the farmers was approximately 45 years with minimum and maximum ages being 18 and 87 years. Some of the farmers had no formal education and some of them completed 22 schooling years, however, most of them were not educated enough. Moreover, some farmers had no previous experience in potato farming. On an average 6 peoples belongs to a potato grower's family. Most of the farmers (72.33%) have no access to credit a loan and about 87.67% of farmers had a chance to use cold storage facilities. Moreover, 80% of the farmers included in this study have no formal training in potato production [Table 1].

**Table 1. Summary statistics of variables used in this study.**

| Variable | Minimum | Maximum | Mean | Standard Deviation |
|---|---|---|---|---|
| Potato production ('000' kg) | 0.28 | 378.00 | 26.39 | 32.24 |
| Land cost ('000' Tk.) | 0.47 | 270.00 | 16.19 | 19.33 |
| Labor cost ('000' Tk.) | 1.20 | 539.00 | 40.99 | 44.51 |
| Tilling cost ('000' Tk.) | 0.43 | 98.00 | 10.41 | 12.36 |
| Seed cost ('000' Tk.) | 1.25 | 940.00 | 41.25 | 66.24 |
| Fertilizer cost ('000' Tk.) | 0.90 | 337.50 | 24.54 | 28.93 |
| Irrigation cost ('000' Tk.) | 0.18 | 54.00 | 3.55 | 4.52 |
| Pesticide cost ('000' Tk.) | 0.10 | 78.57 | 3.63 | 7.01 |
| Vitamin cost ('000' Tk.) | 0.00 | 500.00 | 4.27 | 35.51 |
| Age (Years) | 18 | 87 | 44.81 | 13.59 |
| Education (completed years) | 0 | 22 | 6.30 | 5.24 |
| Experience (years) | 0 | 60 | 11.38 | 10.08 |
| Land fragmentation (number of plots) | 1 | 35 | 4.02 | 3.75 |
| Family size | 2 | 17 | 6.00 | 2.30 |
| Deweed cost ('000' Tk.) | 0.03 | 25.00 | 2.25 | 3.17 |
| Access to credit (Yes, No) | Yes: 83, No: 217 | | | |
| Cold storage (Yes, No) | Yes: 263, No: 37 | | | |
| Training (Yes, No) | Yes: 60, No: 240 | | | |

## Stochastic frontier analysis

Initially, the authors estimate the SFA model using the maximum likelihood method and previous studies also used this methods for estimating parameters [22, 42]. Table 2 shows results of the Cobb-Douglas stochastic frontier model with technical inefficiency factors.

The results show that the coefficients of eight variables are positive and statistically significant at 5% significance level. It indicates that eight variables are important to determine potato production. The elasticity is highest for land. Farmers are not always using their total land for potato cultivation. After the acquisition of land, the land should be prepared well by tilling may be by the cow or by the tractors. The cost of seed positively and significantly influenced potato production and this finding is supported by a previous study [22]. The farmers have to have good quality seeds no matter whether the seeds are local or HYV varieties. Without good quality seeds, production would be hampered since good quality seeds are prerequisites for good production. The farmers have to use good quality fertilizer no matter if they grow local or HYV varieties. Moreover, the lack of high-quality fertilizer will hinder productivity, which is a requirement for increasing production and is consistent with other study findings [22]. It is observed that irrigation; pesticides and vitamins have a relatively small effect [Table 2].

The estimated $\delta$-coefficients of the explanatory variables indicate that the inefficiency variables of farms significantly contribute to explain the technical inefficiency effects in potato farming. The signs of estimated coefficients tell us that these variables cause variation in the technical efficiency of potato farms and affect the capability of farms in adequately utilizing existing technology and Infrastructure. The return to scale $\left( \sum_{i=1}^{8} \beta_i = 2.44 \right)$ indicates increasing returns to scale in potato farming. The estimated value of $\gamma$ parameter is 0.853 which indicates that the farms were inefficient which is in line with other study [22] and the variance parameter $\sigma^2$ is 0.06 in the stochastic frontier. All coefficients of inefficiency factors are statistically significant at 5% level indicating that there are inefficiency effects in the potato farming

**Table 2. Results of Cobb-Douglas stochastic frontier model.**

| Variables | Parameters | Coefficients |
|---|:---:|:---:|
| Constant | $\beta_0$ | 2.799 |
| ln of land | $\beta_1$ | 0.575** |
| ln of labor cost | $\beta_2$ | 0.337** |
| ln of tilling cost | $\beta_3$ | 0.404* |
| ln of seed cost | $\beta_4$ | 0.491* |
| ln of fertilizer cost | $\beta_5$ | 0.285** |
| ln of irrigation cost | $\beta_6$ | 0.152* |
| ln of pesticide cost | $\beta_7$ | 0.126* |
| ln of vitamin cost | $\beta_8$ | 0.068* |
| **Inefficiency Variables** | | |
| Constant | $\delta_0$ | 4.792 |
| Age | $\delta_1$ | -0.524* |
| Education | $\delta_2$ | -0.377** |
| Experience | $\delta_3$ | -0.256** |
| Land fragmentation | $\delta_4$ | 0.338* |
| Family size | $\delta_5$ | 0.097* |
| Deweeding | $\delta_6$ | -0.297* |
| Access to credit (dummy) | $\delta_7$ | -0.383* |
| Cold storage (dummy) | $\delta_8$ | -0.212* |
| Training (dummy) | $\delta_9$ | -0.432* |
| **Variance Parameters** | | |
| $\sigma^2 = \sigma_v^2 + \sigma_u^2$ | | 0.064 |
| $\gamma = \frac{\sigma_u^2}{\sigma^2}$ | | 0.853 |
| $\sigma_v^2$ | | 0.009 |
| $\sigma_u^2$ | | 0.055 |
| Log likelihood | | 324.670 |

Note:

*: p<0.05,

**: 0.05<p<0.01.

in the sample farms and the random factors of the inefficiency effects significantly contribute $\left(\sigma_u^2 = 0.055\right)$ in potato farming efficiency. That is the technical inefficiency effects are important components to determine the variability and level of potato production. The negative coefficients of education, age, and experience imply farmers with more years of schooling, experience, and age are more technically efficient. A previous study highlighted that a farmer's inefficiency will decrease as they get older [43]. Researchers also mentioned that improving the farmer's education level will have a negative impact on their inefficiency [22, 43, 44]. Moreover, previous studies pointed out that growing agricultural experience will have a negative impact on inefficiency [43, 44]. Since weed hampers potato production farmers need to deweed their land for getting more output. As far as land fragmentation is concerned, farmers with smaller plot sizes are more technically inefficient. The higher the smaller plots to be managed it would be difficult to manage the land and especially irrigation and tillage would be difficult with a tractor and water pump with electricity. Since farmers have smaller pieces of land they don't use very much technologically advanced methods of cultivation. Researchers noted that decreasing the size of the land would have a negative impact on technical efficiency scores

[45–48]. The positive coefficient of the family size implies that the higher the family size, the higher the inefficiency though the coefficient is not significant at 5% level.

## Input- oriented DEA frontier

Input-oriented DEA frontiers with CRS and VRS are computed. The ratio of CRS and VRS efficiency estimates the scale efficiency of potato farming. The frequency distribution of farms according to technical and scale efficiencies are presented in S1 Table. It is observed that most of the farms (87%) have technical efficiency scores within (1–70) % efficiency index. Average scores of technical efficiency for CRS and VRS TEs are almost closer. About 83% of farms fall in (70–100) % scale efficiency index. In terms of scale economies, 54% of farms have increasing, 22% have constant and (24%) have decreasing returns to scale [S1 Table]. According to Silberberg and Suen, (2000) farms using the same technology will have increasing return to scale with relatively low output, decreasing return to scale with relatively high output and constant returns to scale when the output level is equal to mean output [49].

## Estimated production, cost and input demand functions

The dual cost frontier is estimated from the stochastic production frontier for the inefficiency components. The stochastic production function and frontier cost function are obtained using the findings presented in Table 2.

Stochastic Production Function is as follows:

$$y_i = 2.799 \, x_{i1}^{0.575} x_{i2}^{0.337} x_{i3}^{0.404} x_{i4}^{0.491} x_{i5}^{0.285} x_{i6}^{0.152} x_{i7}^{0.126} x_{i8}^{0.068},$$

where $i = 1, 2, \ldots, 300$.

Dual cost frontier function is logically estimated as follows:

$$C(p_{ik}, \tilde{y}_i) = 4.435 \, P_{i1}^{0.236} P_{i2}^{0.138} P_{i3}^{0.166} P_{i4}^{0.201} P_{i5}^{0.117} P_{i6}^{0.062} P_{i7}^{0.052} P_{i8}^{0.028} \tilde{y}_i^{0.410},$$

where $i = 1, 2, \ldots, 300$.

Input demand function can be written as

$$x_{i1} = \frac{1.05 p_{i2}^{0.138} p_{i3}^{0.166} p_{i4}^{0.201} p_{i5}^{0.117} p_{i6}^{0.062} p_{i7}^{0.052} p_{i8}^{0.028} \tilde{y}^{0.41}}{p_{i1}^{0.764}},$$

where $i = 1, 2, \ldots, 300$.

## Economic, technical and allocative efficiencies by SFA and DEA method

The frequency distribution (%) of economic, technical, and allocative efficiency scores of farms for both methods are shown in S2 Table. In the SFA method, most of the farms 89% are technologically efficient and falls (70–100%) efficiency class while 11% of farms fall (1–69%) efficiency class. In the case of AE, 68% of farms fall (70–100%) efficiency class, and 32% of farms fall in (1–69%) efficiency class. In the case of EE, 36% of farms fall (70–100%) efficiency class, and 64% of farms fall in (1–69%) efficiency class. Farms have considerable variations in efficiency. The calculated mean of economic, technical, and allocative efficiencies indicates that there are considerable economic, technical, and allocative inefficiencies in potato farming and production can be increased through efficiency improvement. It is seen that TE has the highest mean value and EE has the lowest mean value. Moreover, S2 Table shows the frequency distribution (%) and summary statistics of economic, technical, and allocative efficiencies of CRS and VRS DEA frontier methods. The results of CRS DEA and VRS DEA frontiers reveal that farmers can improve production efficiency without applying more advanced or

new technology in the production process which reduces production costs. Allocative efficiency is greater than the technical and economic efficiencies in potato farming.

## Comparison between the results of SFA and DEA methods

This study has an objective to compare efficiency measures from SFA and DEA methods and to assess whether any significant differences in the estimates of the efficiency. These two methods are different in explaining the gap between the estimated function and observations. To explain the gap SFA method considers both producers' inefficiency and some random elements which are not under the owner's control, but the DEA method supposes the absence of random error and considers only the producer's inefficiency. A previous study pointed out that the empirical result of both SFA and DEA methods, two quite different approaches of efficiency estimation may differ for differences in the choice of input and output variables, characteristics of data analyzed, estimation procedures, and measurement & specification errors [50]. If results are similar, then the measure of efficiency and relative efficiency in terms of socio-economic characteristics and infrastructure factors are robust and can be used as a basis for policy recommendation. For the SFA the efficiency score is 85.3, however, in the case of DEA, technical efficiency for CRS and VRS are 43.76 and 52.71 respectively and scale efficiency is 84.5. In stochastic frontier analysis, it is assumed that all farms have the estimated efficiency scores. But in data envelopment analysis each farm has different efficiency scores. The DEA method represents the average efficiency scores of all farms. From the efficiency score, it is seen that VRS technical efficiency has a higher value which implies that VRS TE enveloped data more closely than CRS TE. The efficiency score of the SFA method is higher than the DEA method.

## Returns to scale

The stochastic frontier analysis estimates returns to scale of farm is $\sum_{i=1}^{8} \beta_i = 2.44$ which is greater than 1. It implies that there is an increasing returns to scale and inefficiency in potato farming. This result is assumed for all farms. Data envelopment analysis estimates returns to scale for all farms separately: 54% of farms have increasing, 22% of farms have constant, and 24% of farms have decreasing return to scale. It is easy to understand which farm has what types of returns to scale. The majority of the farms have increasing returns to scale which has a similar interpretation as SFA that there is inefficiency in potato farming. The following Fig 1 illustrates the mean of economic (EE), technical (TE), and allocative efficiencies (AE) estimates of SFA and DEA methods.

The mean values of efficiency estimates based on Stochastic Frontier Analysis are higher than those based on the CRS and VRS DEA frontier [Fig 1]. It implies that the stochastic frontier is well-fitted to the data set compared to the DEA frontier. Technical efficiency scores of the SFA model are larger than both CRS and VRS DEA models. A previous study also got similar findings for the swine industry in Hawaii [51]. The value of standard deviations shows that the DEA frontier shows larger variability in economic, technical, and allocative efficiency estimates than the stochastic frontier.

## Factors affecting inefficiency from SFA and DEA method

In order to make an appropriate decision, it is crucial to understand the various aspects that affect a company's efficiency. The factors taken into consideration in this research that impact the economic, technical, and allocative efficiencies of SFA and DEA approaches are shown in Table 3.

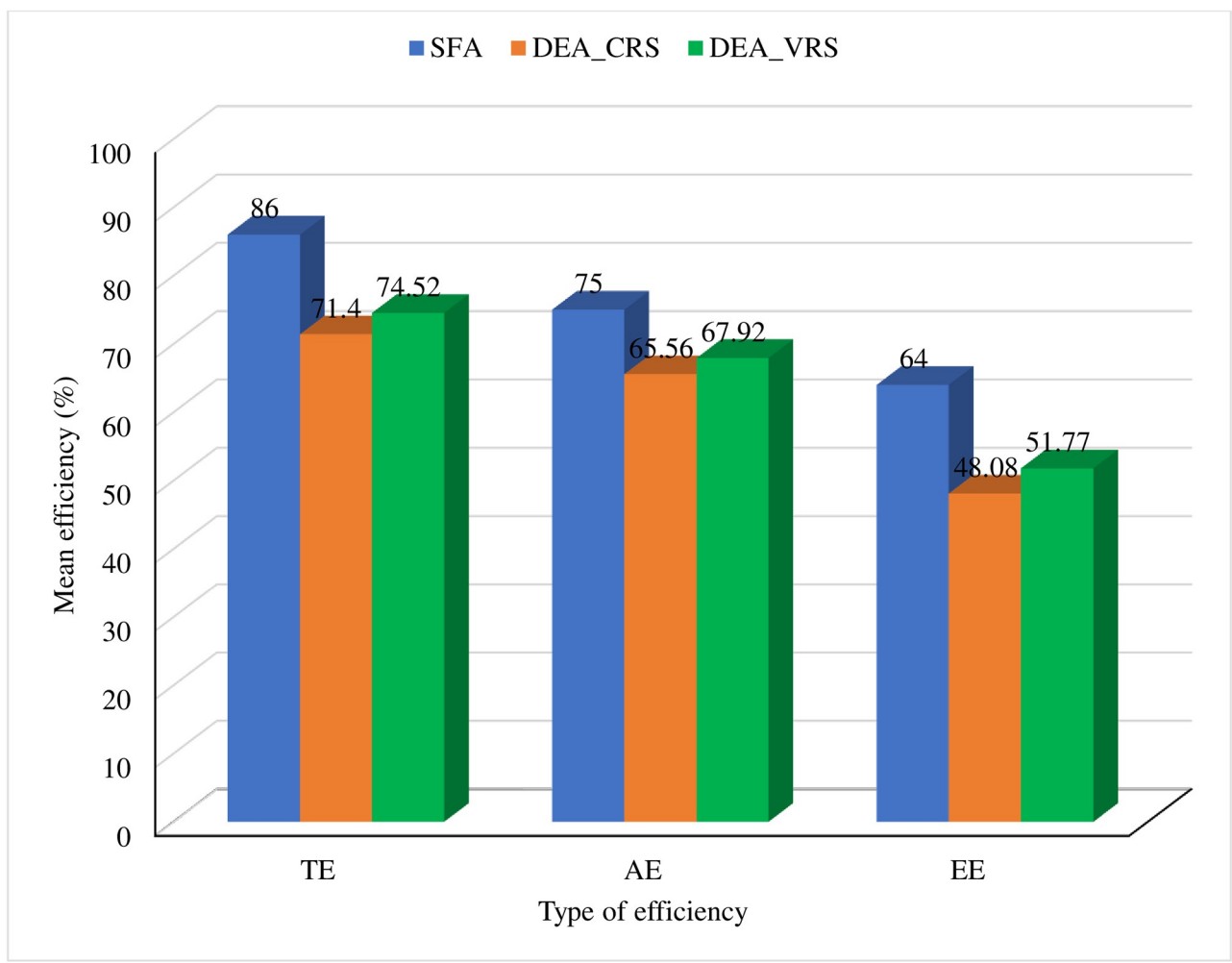

**Fig 1. Mean efficiency of economic, technical and allocative obtained by SFA and DEA methods.**

**Table 3. Coefficients of factors affecting inefficiency.**

| Factors | Stochastic Frontier Analysis (SFA) | | | Data Envelopment Analysis | | | | | |
|---|---|---|---|---|---|---|---|---|---|
| | | | | CRS (Overall TE) | | | VRS (Pure TE) | | |
| | TI | AI | EI | TI | AI | EI | TI | AI | EI |
| Constant | 1.79 | 0.75 | 0.63 | 0.58 | 0.73 | 0.48 | 0.82 | 0.64 | 0.55 |
| Age | -0.52* | -0.05* | -0.01* | -0.03* | -0.01* | -0.02* | -0.05* | -0.01* | -0.01* |
| Education | -0.38* | -0.03* | -0.02* | -0.02* | -0.06* | -0.04* | -0.03* | -0.09* | -0.03* |
| Experience | -0.26* | -0.04* | -0.01* | -0.06* | -0.03* | -0.07* | -0.08* | -0.04* | -0.10* |
| Land fragmentation | 0.34* | 0.02* | 0.03* | 0.02* | 0.01* | 0.01* | 0.06* | 0.06* | 0.02* |
| Family size | 0.097* | 0.009* | 0.006* | 0.012* | 0.002* | 0.009* | 0.027* | 0.002* | 0.022* |
| Deweeding | -0.6* | -0.004* | -0.002* | -0.01* | -0.01* | -0.01* | -0.02* | -0.03* | -0.04* |
| Access to credit (dummy) | -0.98* | -0.02* | -0.02* | -0.04* | -0.09* | -0.04* | -0.07* | -0.08* | -0.11* |
| Cold storage (dummy) | -0.01* | -0.015* | -0.017* | -0.13* | -0.05* | -0.12* | -0.06* | -0.03* | -0.13* |
| Training (dummy) | -1.43* | -0.02* | -0.06* | -0.03* | -0.03* | -0.19* | -0.17* | -0.04* | -0.11* |

*Indicate significant at 5% level

The socio-economic and infrastructural factors' results for both methods are similar in direction and different in coefficient values. Table 3 shows that the coefficient of farming experience is negative for all inefficiencies (TI, AI, EI) denoting that households with more farming experience can improve their skill and capability of managing inputs efficiently and tend to technically efficient potato production. The coefficient for the land fragmentation variable is positive, implying that inefficiency tended to increase with the increase in land fragmentation. If lands are more fragmented, those farms are technically less efficient than those of less fragmented farms. The larger land size is more economic because farmers can easily apply modern technologies like tractors and better irrigation management. The coefficient of family size is positive but not significant, implying that large or small family size has not significantly affected farming efficiency. De-weeding also has a significant effect on potato farming efficiency. The coefficient of education is negative and significant suggests that farmers with more schooling tended to be technically more efficient than farmers with lower education or no education at all.

Two methods give different efficiency scores; provide returns to scale in a different ways. However, from both methods, the authors can understand that potato farming is inefficient. From DEA, it can be identified which farm presents which type of returns to scale. In the case of factors affecting inefficiency, the estimated coefficients are different in value but the same in direction. The efficiency score of the SFA model is higher than the DEA model, which implies that the SFA frontier fits tighter than the DEA frontier. The SFA and VRS DEA are almost similar in the case of estimated technical, allocative, and economic efficiency. In the case of DEA, VRS TE enveloped data more closely than CRS TE. The authors may conclude that Stochastic Frontier Analysis is more applicable to the agriculture sector since the owner cannot control random errors that exist in this sector.

## Limitations of this study

Due to the cross-sectional nature of this investigation, causal inference is not feasible. Due to sample heterogeneity and variability and the fact that the results are based on self-reporting, they can differ in other regional contexts. Furthermore, because this study was self-funded and had a limited budget, it was not able to consider a large sample. As a result, additional research will be conducted with a focus on a representative sample of the entire country.

## Conclusion

Bangladesh achieved commendable success in potato production even under the global context due to favorable agro-climate and lower labor costs. Considering these opportunities, potato farming efficiency can help Bangladesh achieve short- and long-term economic targets. It requires studying the possibility of potato productivity raising through efficiency improvement without introducing new technologies or increasing the resource base. The empirical results of both methods show that the SFA model's efficiency score is higher than the DEA model, which implies that the SFA frontier fits tighter than the DEA frontier. SFA and VRS DEA are almost similar in the case of estimated technical, allocative, and economic efficiency. In the case of DEA, VRS TE enveloped data more closely than CRS TE. The variability and level of production of potato farming are determined by technical inefficiency. The estimated infrastructure and socio-economic factors jointly determine the variabilities of potato production. There are significant economic, technical, and allocative inefficiencies in potato farming and scope to increase potato production levels through efficiency improvement, increasing the farm household's income and welfare. It is concluded that Stochastic Frontier Analysis is more applicable to the agriculture sector because the random error cannot control by the owners in this sector.

This study emphasizes that the following factors need attention to increase potato productivity. Formal education, particularly agriculture-related education, can help farmers increase their knowledge about cultivation and cost-minimizing input use, which can improve allocative efficiency. An extension program could be used to reorient the application of methods, timing, and amount of inputs and production methods. Land tenure and management policies could be designed to reduce land fragmentation that also can lessen obstacles to utilizing existing technology efficiently and helps better allocation of inputs, utilization of irrigation, fertilizer, and land preparation in a cost-minimizing way since there are no training, cold storage, and access to credit facilities. If the farmers have those facilities, the inefficiency of production performance would be reduced substantially. To achieve production efficiency, more emphasis should be placed on choosing the appropriate technology and efficient use of resources for the scale of operation.

## Supporting information

**S1 Table. Frequency distribution (%) of farms according to technical and scale efficiencies.**
(DOCX)

**S2 Table. Frequency distribution (%) of efficiency for farms.**
(DOCX)

**S1 Data. Data set.**
(XLSX)

## Acknowledgments

The authors are grateful to the respondents for participating in this study and providing consent to publish the study findings after removing identifiable information. They are also thankful to the academic editor and two reviewers for their valuable comments and suggestions that helped to enhance the quality of the manuscript.

## Author Contributions

**Conceptualization:** Shamima Sultana, Md. Moyazzem Hossain, Md. Nurul Haque.

**Data curation:** Shamima Sultana.

**Formal analysis:** Shamima Sultana, Md. Moyazzem Hossain.

**Methodology:** Shamima Sultana, Md. Moyazzem Hossain, Md. Nurul Haque.

**Supervision:** Md. Nurul Haque.

**Validation:** Md. Nurul Haque.

**Visualization:** Md. Moyazzem Hossain.

**Writing – original draft:** Shamima Sultana, Md. Moyazzem Hossain.

**Writing – review & editing:** Shamima Sultana, Md. Moyazzem Hossain, Md. Nurul Haque.

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
