## [Decision Letter · Decision Letter 0]

7 Mar 2023

PONE-D-23-04113Estimating the Potato Farming Efficiency: A Comparative Study between Stochastic Frontier Analysis and Data Envelopment AnalysisPLOS ONE

Dear Dr. Hossain,

Thank you for submitting your manuscript to PLOS ONE. After careful consideration, we feel that it has merit but does not fully meet PLOS ONE’s publication criteria as it currently stands. Therefore, we invite you to submit a revised version of the manuscript that addresses the points raised during the review process. Particularly, both reviewers recommend reconsideration of your manuscript following minor revision.  Please submit your revised manuscript by Apr 21 2023 11:59PM. If you will need more time than this to complete your revisions, please reply to this message or contact the journal office at plosone@plos.org. Please include the following items when submitting your revised manuscript:A rebuttal letter that responds to each point raised by the academic editor and reviewer(s). You should upload this letter as a separate file labeled 'Response to Reviewers'.A marked-up copy of your manuscript that highlights changes made to the original version. You should upload this as a separate file labeled 'Revised Manuscript with Track Changes'.An unmarked version of your revised paper without tracked changes. You should upload this as a separate file labeled 'Manuscript'.If applicable, we recommend that you deposit your laboratory protocols in protocols.io to enhance the reproducibility of your results. Protocols.io assigns your protocol its own identifier (DOI) so that it can be cited independently in the future. For instructions see: https://journals.plos.org/plosone/s/submission-guidelines#loc-laboratory-protocols. Additionally, PLOS ONE offers an option for publishing peer-reviewed Lab Protocol articles, which describe protocols hosted on protocols.io. Read more information on sharing protocols at https://plos.org/protocols?utm_medium=editorial-email&utm_source=authorletters&utm_campaign=protocols.

We look forward to receiving your revised manuscript.

Kind regards,

Thanh Ngo, Ph.D.

Academic Editor

PLOS ONE

Journal Requirements:

2. You indicated that ethical approval was not necessary for your study. We understand that the framework for ethical oversight requirements for studies of this type may differ depending on the setting and we would appreciate some further clarification regarding your research. Could you please provide further details on why your study is exempt from the need for approval and confirmation from your institutional review board or research ethics committee (e.g., in the form of a letter or email correspondence) that ethics review was not necessary for this study? Please include a copy of the correspondence as an ""Other"" file.

**Reviewers' comments:**

Reviewer's Responses to Questions

**Comments to the Author**

1. Is the manuscript technically sound, and do the data support the conclusions?

Reviewer #1: Yes

Reviewer #2: Partly

2. Has the statistical analysis been performed appropriately and rigorously? 

Reviewer #1: Yes

Reviewer #2: Yes

3. Have the authors made all data underlying the findings in their manuscript fully available?

Reviewer #1: Yes

Reviewer #2: Yes

4. Is the manuscript presented in an intelligible fashion and written in standard English?

Reviewer #1: Yes

Reviewer #2: No

5. Review Comments to the Author

Reviewer #1: Dear Sir/Madam,

My comments are in the file.

1-The potato (Solanum tuberosum L.) is the most productive crop and the third-largest source of food on earth [1]. (The following information is also available in the literature. Potato is in the fifth place after sugarcane, corn, rice and wheat in the list of the most produced plant products in the world. Please refer to the literature: FAOSTAT (2022). World Production Quantities of Crops. http://www.fao.org/faostat/en/#data/QC. and Kadakoğlu, B., Karlı, B. (2021). Economic Analysis of Potato Production in Afyonkarahisar Province, KSU J. Agric Nat 25 (3): 581-588. https://doi.org/10.18016/ksutarimdoga.vi.947387)

2-To the best of our knowledge, no research has been done yet on potato farming efficiency using both SFA and DEA. (On the contrary, a study was conducted in Türkiye on the technical efficiency of potato cultivation using both SFA and DEA. According to the results of the research, SFA 69.0, DEA-CRS 76.4, DEA-VRS 84.6 were found. Please refer to the literature: Kadakoğlu, B. (2021). Analysis of Technical and Economic Efficiency of Potato Production in Afyonkarahisar Province, Isparta University of Applied Sciences, The Institute of Graduate Education Department of Agricultural Economics, M.Sc. Thesis. http://dx.doi.org/10.13140/RG.2.2.20260.96641)

3-This study applies a self-dual Cobb-Douglas SFA Method. (What is your reason for choosing this method? As you know, there are statistics for Cobb-Douglas or Translog method in SFA and as a result, the method is selected. Please apply the statistics and choose your method accordingly.)

4-Results of Cobb-Douglas stochastic frontier model. (It is recommended to compare your study result with another study on the same method and the same product. Please refer to the literature: Kadakoğlu, B., Karlı, B. (2022). Technical Efficiency of Potato Production in Turkey by Stochastic Frontier Analysis,Custos e Agronegocio, 18(2), 163-178. https://www.researchgate.net/publication/364303318_Technical_Efficiency_of_Potato_Production_in_Turkey_by_Stochastic_Frontier_Analysis)

5-Further research would be better if collected data and information were based on a larger sample size. (Why didn't you increase the sample size of the research more and make it more comprehensive? Include the constraints of your research.)

6-References (It is recommended to include the above-mentioned references to increase the scope of the research.)

Reviewer #2: Estimating the potato farming efficiency: A comparative study between SFA and DEA

Journal: PLOS ONE

Summary

This study first assesses whether SFA is a better method than DEA in estimating potato farming efficiency. Second, this study identifies the determinants of potato farming efficiency.

I only have a few minor comments.

Abstract

Please rewrite the abstract. The current abstract is quite lengthy and repetitive.

Data

Please provide descriptive statistics of variables used in this study.

Please provide references for the inputs and outputs used.

Results

For the analysis of efficiency determinants, please provide references to each finding.

References

Please keep the references consistent.

Other comments

English writing is relatively poor, for example, “where Tk. Is the currency of Bangladesh”. Please approach professional editing services before resubmitting your manuscript. Again, the author(s) could benefit from a professional proofreader.

6. PLOS authors have the option to publish the peer review history of their article (what does this mean?). If published, this will include your full peer review and any attached files.

Reviewer #1: **Yes: **Mevlüt Gül

Reviewer #2: No

---

## [Author Response · Author response to Decision Letter 0]

11 Mar 2023

Dear Editor,

PloS ONE

We would like to express our sincere gratitude to the reviewers and the Editors for their valuable comments. We have considered all the comments made by the reviewers and thoroughly revised and formatted the manuscript accordingly. A detailed response to each of the comments is provided below.

Author responses to the Academic Editor comments:

Thank you very much for carefully checking the manuscript and providing insightful comments. 

All required files are uploaded to the journal system. Revised texts are in red color.

Author responses to the Journal Requirements:

1. Thanks. We revised the manuscript following the PLOS ONE style. Revised texts are in red color.

2. Thanks. We add the consent form as an “Other” file. Revised texts are in red color.

Page: 4

3. Thanks. We add the following. All relevant data are within the manuscript and data are available in a Supporting Information file (S3 Data). Revised texts are in red color.

Page: 18

4. Thanks. We moved it to the Methods section. Revised texts are in red color.

Page: 4

5. Thanks. We add captions of the Supporting Information files at the end of your manuscript. Revised texts are in red color.

Page:17-18

6. Thanks. We checked the reference list and ensure that it is complete and correct. We use Mendeley for citation. 

Authors Response to the Reviewer 1 comments:

1. Thank you very much for carefully checking the manuscript and providing insightful comments. 

We revise it and add the suggested citations. 

Revised texts are in red color. Page: 2 (Ref. 2, 3)

2. Thanks for highlighting this point. 

We have revised the manuscript as per your guidelines. 

Revised texts are in red color. Page: 3 (Ref. 22)

3. We appreciate your comments. We revise the manuscript. 

The Cobb-Douglas type production function has been used extensively by researchers worldwide and the existing literature motivates us to use the self-dual Cobb-Douglas SFA Method in this study. Revised texts are in red color.

Page: 5

4. Thanks. We have revised the Result section according to your guidelines and feedback. Revised texts are in red color.

Page: 7-11

5. Thank you very much. We add a limitation section as per your comments. Revised texts are in red color.

Page: 16

6. We are thankful to you for carefully checking the manuscript. We revised the citation and reference list. Revised texts are in red color.

Page: 18-22

Authors Response to the Reviewer 2 comments:

Thanks. We appreciate your comments. We have revised this section as per your feedback. Revised texts are in red color.

Thank you very much. This section is revised as per the comment. 

We revise the abstract of this manuscript. 

We add the descriptive statistics of variables used in this study.

We add the references for efficiency determinants. 

We use Mendeley for citation and follow the PLOS ONE style. We check and correct all typos and grammatical mistakes. Revised texts are in red color. Page: 1-2, 4, 7-8, 10-11, 12-13

Finally, the revised manuscript has been produced following the valuable comments and suggestions of the reviewers. Once again, we would like to thank the reviewers for their sincere dedication, professional insights, and earnest cooperation in reviewing the manuscript.

---

## [Decision Letter · Decision Letter 1]

27 Mar 2023

PONE-D-23-04113R1Estimating the Potato Farming Efficiency: A Comparative Study between Stochastic Frontier Analysis and Data Envelopment AnalysisPLOS ONE

Dear Dr. Hossain,

Thank you for submitting your manuscript to PLOS ONE. After careful consideration, we feel that it has merit but does not fully meet PLOS ONE’s publication criteria as it currently stands. Therefore, we invite you to submit a revised version of the manuscript that addresses the points raised during the review process.

Besides the comments from Reviewer 2, I suggest the authors to review some missing literature on agriculture efficiency, although not on potato farms, around the references 9 and 10 on page 3 as follows. Fuwa, N., Edmonds, C., & Banik, P. (2007). Are small-scale rice farmers in eastern India really inefficient? Examining the effects of microtopography on technical efficiency estimates. Agricultural Economics, 36(3), 335-346. https://doi.org/10.1111/j.1574-0862.2007.00211.x

 Asadullah, M. N., & Rahman, S. (2009). Farm productivity and efficiency in rural Bangladesh: the role of education revisited. Applied Economics, 41(1), 17-33. https://doi.org/10.1080/00036840601019125

 Madau, F. A. (2011). Parametric Estimation of Technical and Scale Efficiencies in the Italian Citrus Farming. Agricultural Economics Review, 12, 91-111. 
 Madau, F. A., Furesi, R., & Pulina, P. (2017). Technical efficiency and total factor productivity changes in European dairy farm sectors. Agricultural and Food Economics, 5(1), 17. https://doi.org/10.1186/s40100-017-0085-x

 Nguyen, H.-D., Ngo, T., Le, T., Ho, H., & Nguyen, H. T. (2019). The Role of Knowledge in Sustainable Agriculture: Evidence from Rice Farms’ Technical Efficiency in Hanoi, Vietnam. Sustainability, 11(9), 2472. 
 Rada, N. E., & Fuglie, K. O. (2019). New perspectives on farm size and productivity. Food Policy, 84, 147-152. https://doi.org/10.1016/j.foodpol.2018.03.015

We look forward to receiving your revised manuscript.

Kind regards,

Thanh Ngo, Ph.D.

Academic Editor

PLOS ONE

Journal Requirements:

Reviewers' comments:

Reviewer's Responses to Questions

**Comments to the Author**

1. If the authors have adequately addressed your comments raised in a previous round of review and you feel that this manuscript is now acceptable for publication, you may indicate that here to bypass the “Comments to the Author” section, enter your conflict of interest statement in the “Confidential to Editor” section, and submit your "Accept" recommendation.

Reviewer #1: All comments have been addressed

Reviewer #2: (No Response)

2. Is the manuscript technically sound, and do the data support the conclusions?

Reviewer #1: Yes

Reviewer #2: Yes

3. Has the statistical analysis been performed appropriately and rigorously? 

Reviewer #1: Yes

Reviewer #2: Yes

4. Have the authors made all data underlying the findings in their manuscript fully available?

Reviewer #1: Yes

Reviewer #2: Yes

5. Is the manuscript presented in an intelligible fashion and written in standard English?

Reviewer #1: Yes

Reviewer #2: No

6. Review Comments to the Author

Reviewer #1: (No Response)

Reviewer #2: Thank you for giving me a chance to read your work again.

Although the authors have addressed all my comments, I still see several grammar errors there.

For example, pp.10 " The cost on seed is positively and significantly influenced the potato production and this findings is supported by a previous study [16]." or "Moreover, without good quality fertilizer, production would be hampered which is a prerequisite for boosting production which is consistent with other study findings [16]."

In the abstract, "Therefore, the authors aimed to evaluate farm-level efficiency and inefficiency of potato farming in Bangladesh." I think that a word "inefficiency" is unnecessary.

Very best,

Tu

7. PLOS authors have the option to publish the peer review history of their article (what does this mean?). If published, this will include your full peer review and any attached files.

Reviewer #1: **Yes: **Mevlüt Gül

Reviewer #2: No

---

## [Author Response · Author response to Decision Letter 1]

27 Mar 2023

Dear Editor,

We would like to express our sincere gratitude to the reviewers and the Editors for their valuable comments. We have considered all the comments made by the reviewers and thoroughly revised and formatted the manuscript accordingly. A detailed response to each of the comments is provided below.

Author's Response to the Academic Editor comments:

Thank you very much for completing the second round review process and providing feedback. We believe that the suggestion is helpful to improve the quality of the manuscript. 

We appreciate your suggestion. We revised the manuscript in light of the suggested papers and cite them. Revised texts are in red color.

Page: 3, Ref. [11-16]. 

Thank you very much for carefully checking the manuscript and providing insightful comments. 

All required files are uploaded to the journal system. 

Author's Response the Journal Requirements:

Thanks. We checked the reference list and ensure that it is complete and correct. We use Mendeley for citation. 

Author's Response the Reviewer 2 comments:

Thanks. We appreciate your comments. We have revised the manuscript as per your feedback. 

Revised texts are in red color. Page: 1, 10

Finally, the revised manuscript has been produced following the valuable comments and suggestions of the reviewers. Once again, we would like to thank the reviewers for their sincere dedication, professional insights, and earnest cooperation in reviewing the manuscript.

---

## [Editor Report · Decision Letter 2]

29 Mar 2023

Estimating the Potato Farming Efficiency: A Comparative Study between Stochastic Frontier Analysis and Data Envelopment Analysis

PONE-D-23-04113R2

Dear Dr. Hossain,

We’re pleased to inform you that your manuscript has been judged scientifically suitable for publication and will be formally accepted for publication once it meets all outstanding technical requirements.

Kind regards,

Thanh Ngo, Ph.D.

Academic Editor

PLOS ONE
---

## [Editor Report · Acceptance letter]

4 Apr 2023

PONE-D-23-04113R2 

Estimating the Potato Farming Efficiency: A Comparative Study between Stochastic Frontier Analysis and Data Envelopment Analysis 

Dear Dr. Hossain:

I'm pleased to inform you that your manuscript has been deemed suitable for publication in PLOS ONE. Congratulations! Your manuscript is now with our production department. 

Kind regards, 

on behalf of

Dr. Thanh Ngo 

Academic Editor

PLOS ONE